# *Bacillus amyloliquefaciens* TL Downregulates the Ileal Expression of Genes Involved in Immune Responses in Broiler Chickens to Improve Growth Performance

**DOI:** 10.3390/microorganisms9020382

**Published:** 2021-02-13

**Authors:** Yuxuan Hong, Yang Cheng, Leluo Guan, Zutao Zhou, Xiaowen Li, Deshi Shi, Yuncai Xiao

**Affiliations:** 1College of Veterinary Medicine, Huazhong Agricultural University, Wuhan 430070, China; hongyuxuan@163.com (Y.H.); cheng_yang11@126.com (Y.C.); ztzhou@mail.hzau.edu.cn (Z.Z.); misscristine@163.com (X.L.); rock@mail.hzau.edu.cn (D.S.); 2State Key Laboratory of Agricultural Microbiology, Huazhong Agricultural University, Wuhan 430070, China; 3Department of Agricultural, Food & Nutritional Science, University of Alberta, Edmonton, AB T6G 2P5, Canada; lguan@ualberta.ca

**Keywords:** broiler chicken, *Bacillus amyloliquefaciens* TL, differentially expressed gene, transcriptomics

## Abstract

*Bacillus amyloliquefaciens* TL promotes broiler chicken performance by improving nutrient absorption and utilization and reducing intestinal inflammation. In this study, RNA-sequencing (RNA-seq)-based transcriptomes of ileal tissues collected from probiotic-fed and control broiler chickens were analyzed to elucidate the effects of the probiotic *B. amyloliquefaciens* TL, as a feed additive, on the gut immune function. In total, 475 genes were significantly differentially expressed between the ileum of probiotic-fed and control birds. The expression of genes encoding pyruvate kinase, prothymosin-α, and heat stress proteins was high in the ileum of probiotic-fed birds (FPKM > 500), but not in the control group. The gene ontology functional enrichment and pathway enrichment analyses revealed that the uniquely expressed genes in the control group were mostly involved in immune responses, whereas those in the probiotic group were involved in fibroblast growth factor receptor signaling pathways and positive regulation of cell proliferation. *Bacillus amyloliquefaciens* TL downregulated the expression of certain proinflammatory factors and affected the cytokine–cytokine receptor interaction pathway. Furthermore, *B. amyloliquefaciens* TL in broiler diets altered the expression of genes involved in immune functions in the ileum. Thus, it might contribute to improved broiler growth by regulating the immune system and reducing intestinal damage in broilers.

## 1. Introduction

Over the past 20 years, the broiler chicken industry has made substantial advances to improve bird productivity through breeding and nutrition management. Various feed additives such as antimicrobials, probiotics, and prebiotics have been widely applied to improve the growth performance of birds [1]. Among them, probiotics have been shown to improve immunity, gut function, health, and growth parameters in broiler chickens [2,3]. Moreover, supplementing the diet with probiotics has been reported to enhance nutrient digestibility and improve the cecal microflora composition in broiler chickens [4]. It has been demonstrated that dietary supplementation of *Bacillus* spp. can alter the intestinal microbiota to help prevent infectious diseases and increase productivity [5,6,7,8,9]. It has also been shown that some *Bacillus subtilis* strains can promote growth in chickens [10]. However, the effects of probiotics on gastrointestinal tract physiology and function in chickens are still poorly understood.

In birds, the intestine, especially the small intestine, is the main site of absorption and metabolism of nutrients derived from feed; it is also a primary site for pathogen entry [11,12]. It has important immunological, endocrine, and regulatory functions [13] that substantially affect animal health [12,14]. The ileum, the longest segment of the small intestine, harbors the majority of gut-associated lymphoid tissue [15], has a higher abundance of microbiota and stronger fermenting ability than the duodenum and jejunum, and it produces various microbial metabolites [16,17]. The gut microbiome can affect host animal health in several aspects, including the provision of nutrients and vitamins, protection against pathogens, development of the immune system, and homeostasis of epithelial mucosa [18]. Therefore, the establishment and maintenance of beneficial interactions between the host and its associated gut microbiota is a key health requirement [19]. To date, most studies have mainly focused on the cecum and reported that probiotics can inhibit the negative effects of intestinal disorders in broiler chickens by reducing cecal pathogens and regulating stress reactions [20,21]. However, there is still a gap in our understanding regarding the ileum and its functional changes in response to probiotics in chicken.

Studies have revealed the effects of immunity on broiler growth performance [22,23,24]. Klasing and Austic showed that immune system stimulation leads to a decrease in the growth performance of growing chicks [25]. In general, immune system activation is energetically costly, and long-term stimulation can have negative effects on the host [24]. Our previous study showed that dietary supplementation with *Bacillus amyloliquefaciens* TL improves the growth of broilers [26]. We speculated that this phenomenon was due to *B. amyloliquefaciens* TL, which can alter the gut microbiota and subsequently lower intestinal inflammation and the systemic immune response, diverting more energy toward growth. Therefore, in this study, a transcriptome analysis of the ileum of broiler chickens was performed to elucidate the functional changes in this tissue in response to the probiotic *B. amyloliquefaciens* TL. This study will provide a theoretical basis for future studies on the regulatory and molecular mechanisms underlying the effects of probiotics on animal performance.

## 2. Materials and Methods

### 2.1. Laboratory Animals

The present study was performed in strict accordance with the Guide for the Care and Use of Laboratory Animals Monitoring Committee of Hubei Province, China, and the protocol was approved by the Committee on the Ethics of Animal Experiments of the College of Veterinary Medicine, Huazhong Agricultural University (NO. HZAUCH-2017-012). For this study, chickens were reared in cages. In total, 120 1-day-old male Cobb broiler chickens were randomly divided into two groups—control and probiotic—with each group consisting of four pens and 15 chickens per pen (n = 60 chickens per group in a total of 120). The control group chickens were fed a basal diet (no drugs or additives), and the probiotic group chickens were fed grain with *B. amyloliquefaciens* TL 4 × 10^6^ cfu/g) from 1 to 35 days of age. The *B. amyloliquefaciens* TL strain was obtained from Hubei Huada Real Technology Co. (Wuhan, China). *B. amyloliquefaciens* TL were added to the feed in the form of dry powder at a concentration of 200 g/ton, with an effective viable number of 2.0 × 10^10^ cfu/g; the dose was advised by the product description. Previous studies in this laboratory have shown that the effect of probiotics at a concentration of 4×10^6^ cfu/g is similar to the feed additive chlortetracycline at 50 g/ton [26].

The chicken coop (which housed all pens) and the surrounding areas were disinfected with potassium permanganate and formalin before the trial. The temperature of the chicken coop was maintained at approximately 33 °C until the chickens were 7 days old; then, it was gradually reduced to 23 °C when the chickens were between 7 and 21 days of age and maintained at 23 °C thereafter. To keep the chickens healthy, the coop was cleared of manure daily, and the chickens had access to free ventilation; water and feed were provided ad libitum. Feed consumption by chickens in both groups and residual feed were recorded daily. Body weights of the chickens were measured on days 1, 7, 14, 21, and 35 (n = 60 chickens per group in a total of 120). During the experiment, the control group and the probiotic group each had two chickens die over 1–7 days due to individual weakness. Thereafter, the weights of the dead broilers and the remaining feed weight were recorded to correct the growth performance data.

### 2.2. Tissue Sampling

Six chickens were sampled from the control and probiotic groups on day 21 (n = 6 samples per group in a total of 12 samples), and the pen number and weight of each individual before slaughter were recorded. The method of selecting samples is: select the individuals in the four pens that are closest to the average weight of each pen. In order to expand the biological replicate sample, we selected the two closest to the average weight of each group from all individuals in each group. We selected the most representative individuals for sampling as much as possible to sequence the gut transcriptome and used these representative data to predict the entire population. The basic information of the sampled individuals is shown in Appendix A. Following their respective fasting periods, the birds were euthanized by cervical dislocation, and the intestinal tissues were collected for further analyses. All tissues were washed with ice-cold phosphate buffer solution (Sigma-Aldrich, St. Louis, MO, USA), snap-frozen in liquid nitrogen, and stored at −80 °C until mRNA expression analysis.

### 2.3. RNA Preparation and Sequencing

The total RNA was extracted from each ileal sample using the RNATM.iso PLUS Kit (Takara Biotechnology, Shiga, Japan). RNA degradation and purity were assessed by 1% agarose gel electrophoresis, and purity was further examined using a NanoPhotometer^®^ spectrophotometer (Implen, Westlake Village, CA, USA) [27]. RNA was quantified using the Qubit^®^ RNA Assay Kit and Qubit^®^ 2.0 Fluorometer (Life Technologies, San Diego, CA, USA) [27]. RNA integrity was assessed using the RNA Nano 6000 Assay Kit with a Bioanalyzer 2100 system (Agilent Technologies, Santa Clara, CA, USA) [27]. The samples that passed quality control standards (specifically, the target band of RNA electrophoresis gel is bright and clear, there is no diffusion area in the swimming lane, and no protein and DNA contamination; RNA Integrity Number (RIN) value is close to 10; 28S/18S is greater than or equal to 1.5; 1.7 < OD260/280 < 2.0; OD260/230 ratio is 2.5) were submitted to Novogene Co., Ltd. (Beijing, China) for library preparation and sequencing using the Illumina HiSeq platform, and 125–/150 bp paired-end reads were generated (n = 6 samples per group in a total of 12 samples) [28].

### 2.4. Quality Control

Transcriptome assembly and annotation protocols were provided by Novogene Co., Ltd. (Beijing, China). To obtain high-quality sequences, the raw reads were filtered by removing the following: (1) reads containing adapters; (2) reads containing poly-N; (3) low-quality reads [29]. Once the clean data were obtained, Q20, Q30, and GC content were calculated. The subsequent analyses were performed using the high-quality clean data of [27].

### 2.5. Mapping Reads to the Reference Genome

Reference genome and gene model annotation files were downloaded directly from the genome website (https://www.ncbi.nlm.nih.gov (accessed on 13 February 2021) [10/2017]). An index was built for the reference genome using Bowtie v2.2.3, and paired-end clean reads were aligned to the reference genome using TopHat v2.0.12 [27]. We selected TopHat as the mapping tool because it can generate a database of splice junctions based on the gene model annotation file, yielding the best mapping results compared to all nonsplice mapping tools [30].

### 2.6. Quantification of Gene Expression

HTSeq v0.6.1 was used to count the read numbers mapped to each gene. The fragments per kilobase of transcript per million fragments mapped (FPKM) of each gene were then calculated based on the length of the gene and the number of reads mapped to it. The expected FPKM number simultaneously considers the effect of sequencing depth and gene length on the number of reads and is currently the most commonly used method for estimating gene expression levels [31].

### 2.7. Differential Expression Analysis

Differential expression analysis was performed based on two conditions/groups (two biological replicates per condition) using the DESeq R package (1.18.0). DESeq provides statistical routines for detecting differential expression in the digital gene expression data using a model that is based on a negative binomial distribution. The resulting *p*-values were adjusted using Benjamini and Hochberg’s approach for controlling the false discovery rate [32]. Genes with adjusted *p*-values < 0.05 were detected by DESeq and were assigned as differentially expressed fragments (i.e., per kilobase of transcript sequence per million base pairs sequenced).

### 2.8. Gene Ontology (GO) and Pathway Enrichment

Based on the DEG data after screening, pathway enrichment with the Kyoto Encyclopedia of Genes and Genomes (KEGG) and GO was performed using DAVID (https://david.ncifcrf.gov/ (accessed on 13 February 2021) [10/2017]), with *Gallus gallus* as the reference.

### 2.9. Quantitative Real-Time Polymerase Chain Reaction Analysis

For quantitative real-time polymerase chain reaction (qRT-PCR) validation experiments, six genes were randomly selected to assess the RNA-sequencing (RNA-Seq) data (Table 1). For this analysis, 1 μg of RNA was reverse-transcribed into cDNA using the PrimeScript™ RT Reagent Kit with gDNA Eraser (TaKaRa, Dalian, China) according to the manufacturer’s instructions. cDNA was diluted 10-fold and used for real-time PCR analyses with a Bio-Rad CFX96TM System and signal detection protocols in accordance with the manufacturer’s instructions (Takara). qRT-PCR experiments were performed in three technical replicates. *β-Actin* was used as the endogenous control. The expression of individual genes was normalized to that of *β-Actin*, a house-keeping gene [33]. Primers used for qRT-PCR are shown in Table 1. Data were analyzed using GraphPad Prism v 5.0 Software (GraphPad, Inc., La Jolla, CA, USA).

### 2.10. Statistical Analyses and Data Records

Statistical evaluations were performed using SPSS for Windows, version 22 (SPSS, Inc., Chicago, IL, USA). Graphs were plotted using GraphPad Prism 5 software (GraphPad, Inc.). The results are presented as the mean ± SEM. The significance levels for all analyses were set as *p* < 0.05 (*) and *p* < 0.01 (**).

## 3. Results

### 3.1. Growth Performance

The average daily gain (ADG) and overall feed conversion ratio (FCR) are shown in Table 2. The average daily gain of the probiotic group was significantly higher than that of the control group at 0–7 (*p* < 0.05) and 14–21 days of age, and the difference was the most significant at 14–21 days (*p* < 0.01). At 14–21 days of age, the average weight and average daily gain of broilers in the probiotic group were 25.33% and 51.47% higher than those in the control group, respectively. Additionally, the overall FCR of broilers in the probiotic group was lower than that in the control group (Table 2).

### 3.2. Transcriptome Analysis

The transcriptome data from the 12 ileum samples are presented in Appendix A (n = 6 samples per group in a total of 12 samples). After removing the low-quality data, 6.83–10.38 Gb of clean data were obtained for further analyses, with >88% of the filtered reads having a Phred quality score of >30 (base Q30). As shown in Figure 1a, the Pearson correlation (R^2^) of the biological repeats in the control and probiotic groups was high, indicating that the sequencing data were suitable for subsequent analyses. The transcriptome profiles of the ileal tissues from the probiotic group were separated from those of the control group in the Principal Component Analysis (PCA) plot (Figure 1b). The expression of 14,033 and 14,446 genes was detected in the ileal tissues of birds from the control and probiotic groups, respectively, with 13,759 genes expressed in both groups (defined as the core transcriptome) (Figure 1c), based on the FPKM, which was >1 and was detected in >50% of the samples in each group.

### 3.3. Functional Analysis of Broiler Chicken Transcriptomes

We performed a functional enrichment analysis of all genes expressed in the ileal tissues of broilers in the two groups (n = 6 samples per group in a total of 12 samples). The top 60 functionally enriched GO terms (the top 20 terms each in the biological process (BP), cellular component (CC), and molecular function (MF) categories) in the ileal tissue of the control and probiotic groups are shown in Figure 2a,b, respectively. Among them, 54 GO terms were enriched in both groups, six were enriched in the control group only, and six were enriched only in the probiotic group (Figure 2c and Appendix A). In addition to normal cellular activities such as “cell division”, “DNA replication”, and “intracellular protein transport”, the GO terms related to intestinal function that were enriched in both groups included “positive regulation of I-kappaB kinase/NF-kappaB signaling”, “cellular response to mechanical stimulus”, “positive regulation of defense response to virus by host”, and “xenophagy”.

We performed functional enrichment analysis of the uniquely expressed genes in the ileal tissues of the two groups of broilers (n = 6 samples per group in a total of 12 samples). In total, 274 genes were only expressed in the control group and 687 were expressed only in the probiotic group. The following GO terms were enriched for gene function in the control group: “immune response”, “chemotaxis”, “regulation of inflammatory response”, “thrombin receptor signaling pathway”, and “somatic hypermutation of immunoglobulin genes” (Figure 3a). The following GO terms were enriched in the uniquely expressed genes of the probiotic group: “fibroblast growth factor receptor signaling pathway”, “positive regulation of cell proliferation”, “integral component of plasma membrane”, and “G-protein coupled serotonin receptor activity” (Figure 3b). The significantly enriched KEGG pathways involving the uniquely expressed genes in the control group included “intestinal immune network for IgA production” and “cytokine–cytokine receptor interaction” (Figure 3c). The uniquely expressed genes in the probiotic group were mainly enriched in “regulation of actin cytoskeleton” and “neuroactive ligand–receptor interaction” (Figure 3d). There were 243 and 229 highly expressed genes (FPKM > 500) in the control and probiotic groups, respectively (Appendix A). Genes encoding pyruvate kinase M (PKM), prothymosin-α, and heat stress proteins (HSPs) were highly expressed in the ileum of the probiotic group but not in that the control group (Appendix A).

### 3.4. Differentially Expressed Genes in the Ileal Tissue and Their Functions

In total, 475 genes were found to be differentially expressed in the ileum (*p* < 0.05) (n = 6 samples per group in a total of 12 samples) with 321 upregulated (67.58%) and 154 downregulated genes (32.42%) in the probiotic group based on the cut-off for differentially expressed genes (DEGs) with *p*-adj < 0.05 and detected in >50% of samples within each group (Appendix A). The GO enrichment analysis showed that the functions of DEGs could be classified as BP, CC, and MF categories. The enriched GO terms of the DEGs are shown in Figure 4a; a high proportion of the DEGs upregulated in the probiotic group were involved in BPs including “regulation of cell migration”, “multicellular organism development”, “neural crest cell migration”, and “coronary vasculature development” categories (Figure 4a and Appendix A). The enriched GO terms of the downregulated DEGs in the probiotic group were in the “establishment of T cell polarity” and “immunoglobulin production” categories (Figure 4b and Appendix A). In addition, KEGG pathways were used to predict gene functions. The pathway analysis of downregulated DEGs between the two groups highlighted “cytokine–cytokine receptor interaction” as the most significantly influenced pathway. The upregulated DEGs were significantly enriched in the “calcium signaling pathway”, “focal adhesion”, and “insulin signaling pathway” (Table 3 and Figure 4c,d).

### 3.5. Confirmation of Gene Expression Data by Quantitative Reverse Transcription Polymerase Chain Reaction

To validate the identified ileal DEGs between the control and probiotic groups determined by RNA-Seq, six DEGs (*CXCR5*, *CXCL13*, *IL-22*, *ACACB*, *LEPR*, and *LAMA5*) were selected for qRT-PCR analysis. The results of high-throughput RNA-seq revealed that *ACACB*, *LEPR*, and *LAMA5* were upregulated, whereas *CXCR5*, *CXCL13*, and *IL-22* were downregulated. The qRT-PCR results showed similar trends of upregulation and downregulation (Figure 5), validating the high-throughput RNA-seq results.

## 4. Discussion

Our study showed that poultry feed supplemented with *B. amyloliquefaciens* TL significantly increased the weight of broilers at 21 days (*p* < 0.05) and reduced the feed:gain ratio. The results indicate that *B. amyloliquefaciens* TL as a probiotic promotes growth and improves feed efficiency. A previous study has shown that although dietary supplementation with the probiotic *B. subtilis* does not have any effect on broiler growth performance between 1 to 21 days, broilers fed probiotics presented higher ADG and average daily feed intake and a lower FCR than broilers with no probiotic supplementation (*p* < 0.05) at 22–42 days [34]. Furthermore, Willie et al. reported that broiler chicks supplemented with commercial probiotic cultures (*Lactobacillus acidophilus*, *Lactobacillus casei*, *Bifidobacterium*, and *Enterococcus faecalis*) showed the highest weight gain of 0.62 kg at 3 weeks [35]. Furthermore, the addition of probiotics (*L. acidophilus*, *B. subtilis*, and *Clostridium butyricum*) to broiler diets resulted in the highest weight gain at 22–35 days, a 7.91% increase over that of the control group (*p* < 0.05) [36]. Although performance improvement varied in these trials, it is evident that probiotic supplementation can lead to higher productivity of broilers at different growth stages. However, these studies did not use *B. amyloliquefaciens* TL, and its growth-promoting effect in broilers was particularly evident at 14–21 days of age, which is different from the results of previous studies. The present study results suggest that the growth-promoting effect of *B. amyloliquefaciens* TL is better during the broiler brooding period (1–21 days) than the fattening period (22–35 days).

This is the first study to report the transcriptome and its functions in the ileum of broiler chickens at 21 days of age and identify whether *B. amyloliquefaciens* TL could affect tissue function at the molecular level. The transcriptome profiles of the ileal mucosal genes in two fast-growing chicken hybrids (HA and HB) at 43 days have been reported [12]. The body weight, daily weight gain, and daily feed intake of HB broilers were higher than those of the HA broilers, but HA broilers showed earlier development characteristics. The ileal mucosal transcriptome of Zampiga et al. showed that a high percentage of gene sets involved in cell energy metabolism, mitochondria structure and functionality, ribosome structure, protein synthesis, cell structure and integrity, and antioxidant and detox mechanisms could be observed in the HA group. Gene sets involved in immune system activation, signal transduction and cell signal transduction, DNA remodeling and replication-chromatin/histone modification, cell activation, migration and adhesion, inflammation, and bone remodeling were detected in the HB group [12]. Here, the functional analysis of the core transcriptome revealed that in addition to some normal cellular activities such as cell division and protein transport, which were similar to the results of Zampiga et al., the following functions were enriched: “positive regulation of I-kappaB kinase/NF-kappaB signaling”, “cellular response to mechanical stimulus”, “positive regulation of defense response to virus by host”, and “xenophagy”. This indicates that the ileum is involved in positively regulating the activation of the immune system to resist bacterial and viral invasions in the intestine. Xenophagy is a selective form of autophagy and is also specifically used to inhibit cellular activities of intracellular bacteria [37]. These results concur with the transcription profile of HB broilers [12]. The ileum transcriptome is involved in the regulation of some immune signaling pathways and defense responses to viruses and bacteria, suggesting that these immune functions are among the core functions of the chicken ileum.

In the present study, the uniquely expressed genes in the ileal tissue of the control group were mostly related to immune responses (e.g., *CD28*, *CXCR5*, *VPREB3*, and *CCR6*), with "immune response" and "chemotaxis" GO terms and "intestinal immune network for IgA production" and "cytokine–cytokine receptor interaction" pathways. CD28 is a secondary signaling receptor that activates T cells by interacting with costimulatory molecules on antigen-presenting cells, such as CD80/86 [38]. In chickens, CD28 is expressed in most αβ T cells, and it has similar functional properties to mammalian CD28 [39]. CCR6 is a specific receptor for CCL20, and both have been cloned and expressed in chickens. Chicken CCR6 is mainly expressed in bone marrow, secondary lymphoid organs, and on the mucosal surface [40]. The expression of chicken *CCR6* suggests immature dendritic cells [41]. Chicken VpreB3 is expressed in immature and/or mature IgM-positive cells, binds to free IgLC, and negatively regulates its maturation and secretion [42]. Studies have shown that after the injection of lipopolysaccharides, the *CXCR5* level in the bursa decreases, and the *CXCR5* mRNA level in the cecum tonsil increases. Chicken *CXCR5* might be involved in the migration of bursal immune cells [43]. Here, these genes were expressed only in the control group, indicating that the immune activity of the intestines of 21-day-old broilers is more active in the control than in the probiotic group. Studies have shown that probiotics can effectively reduce the risk of intestinal infection and colonization of multiple pathogens [44,45]. We speculate that in the ileum of birds receiving *B. amyloliquefaciens*, there could be fewer pathogens. However, future studies are needed to quantify the pathogen populations between the two groups to support such speculations.

Here, *FGF19*, *FGF16*, *FGF10*, *TRIM71*, and *AREG* were only expressed in the probiotic group, and they are mainly associated with the GO terms “fibroblast growth factor receptor signaling pathway” and “positive regulation of cell proliferation.” Fibroblast growth factors (FGFs) are signaling proteins that were originally discovered as stimuli for the growth of fibroblasts or epithelial cells and have multiple functions in development, metabolism, and nerve function [46]. Chicken FGF10 is expressed predominantly in abdominal fat and is involved in preadipocyte differentiation and adipocyte maturation [47]. FGF19 is mainly secreted by ileal enterocytes in response to bile acid activation of the nuclear receptor FXR, and then, FGF19 acts on the cell surface receptor complex in hepatocytes to inhibit bile acid synthesis and gluconeogenesis and stimulate glycogen protein synthesis [48]. Studies on rodents claim that FGF16 also has a regulatory effect on bile acid metabolism [49]. However, in birds, the role of FGF19 in retinal development and that of FGF16 in inner ear development have been reported [50,51], whereas research on the functions of FGF19 and FGF16 in the gut is relatively limited. Considering these findings, the unique expression of the three genes of the FGF family in the probiotic group suggests that glycogen and protein syntheses and fat differentiation were more active in these broilers. AREG is a ligand for epidermal growth factor receptor (EGFR) and is involved in a wide range of physiological processes, including breast development, blastocyst implantation, keratinocyte proliferation, nerve and bone tissue development, and axon growth [52,53]. TRIM71 belongs to the TRIM-NHL protein family. The TRIM proteins control important cellular processes, such as intracellular signaling, innate immunity, transcription, autophagy, carcinogenesis, and antiretroviral action. For example, TRIM22 can inhibit human immunodeficiency virus 1 long-term and repeat-driven transcription according to the RING domain, and TRIM62 plays an important role in restricting the replication of avian leukemia virus subgroup J. In this study, *TRIM71* was uniquely expressed in the probiotic group. TRIM71 plays a conservative role in regulating early development and differentiation and has the function of inhibiting tumorigenesis [54], but there is no related function in birds. The unique expression of these genes in the probiotic group indicates that this group has higher disease resistance, which would enable the rapid growth of broilers at this stage.

The ileal transcriptome showed that the *PKM*, *prothymosin-α*, and *HSPA8* genes were highly expressed in the probiotic group, but not in the control group. PKM activation might increase glucose metabolism flux by catalyzing the conversion of phosphoenolpyruvate to ATP and pyruvate, which is used as a raw material for synthesizing triglycerides from acetyl-CoA [55]. Therefore, the higher *PKM* expression in the ileal tissues of the probiotic group indicates that the broilers in this group have higher metabolism and energy requirements. Prothymosin-α is an immunomodulator with antiviral, antifungal, and anticancer functions, with several features similar to IL-1α [56]. The high expression of *prothymosin-α* in the probiotic group might lead to improved immune function and disease resistance, which could be one of the reasons for the better growth of *B. amyloliquefaciens* TL-fed chickens. *HSPA8* encodes a member of the heat shock protein 70 family, which plays a major role in maintaining a stable cellular environment under high heat stress conditions in animals [57,58]. One study showed that *Hsp70* inhibition enhances heat stress injury and apoptosis in chicken primary cardiomyocytes, and *Hsp70* has a cytoprotective effect during heat stress in chickens [59]. In addition, the HSP70 family has an immunomodulatory effect, which can suppress the immune phenotype by abrogating immune cell interactions, thereby inhibiting the inflammatory pathway [58]. The high expression of *HSP70* in the probiotic group also indicates that feeding broilers the probiotic *B. amyloliquefaciens* TL enables them to respond more effectively to heat stress by regulating immunity. We did not perform a heat stress challenge in this study, and this conclusion will be verified by subsequent experiments.

Among the DEGs, the immune-related DEGs (such as *IL-8*, *CXCL13L2*, *CXCR5*, *TNFRSF13C*, and *IFNGR1*) showed significantly different expressions. *IL-8*, a downregulated DEG in the probiotic group, is an important immunomodulatory factor. IL-8 was the first *CXC* chemokine identified in chickens [60], and it plays an important role in the mucosa by attracting chemokines and neutrophils to cause an inflammatory response [61]. *IL-8* mRNA was shown to increase in the intestinal tissues and livers of birds after infection with *Salmonella* [62]. According to Ateya et al., supplementing probiotics, acidifiers, and symbiotics to the diet of broilers infected with *Escherichia coli* can reduce the expression of ileal *IL-8*, which regulates intestinal inflammation in response to *E. coli* infection and minimizes inflammation-induced damage, thereby improving growth performance [63]. Here, the downregulation of *IL-8* expression in the ileum of the probiotic group indicates that the supplementation with TL reduces chronic inflammation of the intestine. It might also reduce energy consumption caused by intestinal inflammation pressure.

CXCL13 and its receptor CXCR5 (which was downregulated in the probiotic group) also play important roles in infection, inflammation, and the immune response [64]. Studies have shown that chicken CXCR5 is expressed on the surface of all B cell and T cell subpopulations, and CXCL13 effectively binds to cells expressing CXCR5. CXCL13 is a B cell chemoattractant and plays an important role in lymphoid neoplasia. It is also extensively involved in the pathogenesis of several autoimmune diseases, inflammatory conditions, and lymphoproliferative disorders [65]. The chemokine-receptor pair (CXCL13-CXCR5) is also conserved between mammalian and avian species [66]. Considering the important role of this complex in inflammatory diseases, its downregulation indicates a lower level of ileal inflammation in broiler chickens in the probiotic group.

*TNFRSF13C* encodes the BAFF receptor (BAFFR). Chicken BAFF is synthesized by bursal stromal cells or immature B cells [67]. It is expressed mainly in the bursa of Fabricius. The receptor has also been detected in peripheral B lymphocytes in organs such as the cecal tonsil, and it can bind to BAFF [68]. BAFF treatment increases the number of B cells in the cecal tonsil and the expression of BAFFR [68]. Chicken BAFF and BAFFR play important roles in early B cell survival and mature B cell proliferation in chickens [68]. Downregulation of *TNFRSF13C* indicates that *B. amyloliquefaciens* TL might reduce peripheral B lymphocyte numbers in the ileum and intestine-specific immune response.

Avian IFNGR1 is one of the receptors of IFN-γ, which is mainly expressed in chicken lymphoid tissues and organs; it is also expressed in tissue cells such as the small intestine and breast muscle cells [69]. In birds, IFN-γ is a key cytokine for I-type T helper (Th1) cell responses and is essential for controlling infection by intracellular pathogens [70]. Avian IFN-γ can be produced by macrophages, dendritic cells, and CD4^+^ and CD8^+^ T cells, as in mammals [71]. It has been reported that IFN-γ in mammals can prevent the infiltration and destruction of virions in cells during virus infection, as well as the formation of virions or germination of viruses. It can further stimulate cells of the immune system to protect the human body [72]. Kidane et al. reported that in vitro attenuated *Histomonas meleagridis* vaccination results in high levels of IFN-γ in the cecum of turkeys, which might involve important mechanisms of immune responses against avian diseases [71]. In the present study, the decrease in *IFNGR1* expression can be attributed to the lower stimulation of the ileal immune system by viruses and pathogens and a lower Th1 immune response in the ileum of broilers in the probiotic group. Similar studies have shown that a decrease in IFN-γ production by spleen cells in *Lactobacillus*-fed chickens reflects a selective decrease in Th1 cell activation [73].

CCL19 could act as a lymphocyte chemoattractant factor via CCR7 in some virus infections or inflammatory disorders [74]. For example, studies have shown that CCL19 played an important role in T cell migration into bursae during IBDV infection [75,76]. In short, the role of CCL19 in chickens is summarized as follows: on the one hand, the main role of CCL19 is to recruit immune cells into the lesions to limit virus infection and inflammatory responses; on the other hand, CCL19 could facilitate virus invasion [76]. The downregulation of the CCL19 gene in the ileum of broiler chickens in the probiotic treatment group showed that TL reduced the chemotaxis of T cells, thereby reducing the energy consumption of the immune response.

The GO analysis of DEGs revealed that the downregulated genes were predominantly involved in the “establishment of T cell polarity” and “immunoglobulin production”. The KEGG analysis of the downregulated genes indicated that they were mainly involved in the “cytokine–cytokine receptor interaction”, and most DEGs enriched in this pathway are related to the immune response. Figure 6 shows the proposed regulatory and molecular mechanisms of *B. amyloliquefaciens* TL. We conclude that the effect of *B. amyloliquefaciens* TL mainly involves the downregulation of the expression of some immune-related genes in the ileum. In the probiotic group, the expression of some important inflammatory factors and their receptors at 21 days were lower than those in the control group, thereby reducing the activity of immune cells and the production of immunoglobulins. This indicates the following benefits of supplementing probiotics in the feed: a reduction in intestinal inflammation levels and intestinal damage.

In recent years, increasing attention has been paid to the use of probiotics as an alternative feed additive to antibiotics. Previous studies have confirmed the ability of probiotics to enhance the gut microbial balance and consequently the natural defense system of animals against pathogenic bacteria. During production, broilers are susceptible to environmental and pathogenic stresses, which are detrimental to growth. Probiotics act as potential immune modulators and protect these birds during growth. Gene expression analysis in the cecal tonsils of chickens treated with probiotics and challenged with *Salmonella* indicated that probiotics inhibit IL-12 and IFN-γ secretion [77]. In the present study, the reduced immune activity of the probiotic group might have assisted broilers in diverting nutritional energy to promote body growth. Excessive immune stress in broilers can affect growth performance. Therefore, the industry should avoid unnecessary innate immune activity, such as chronic inflammation. The probiotic *B. amyloliquefaciens* TL can reduce immune stress to some extent, which explains the weight gain in our probiotic group. The present study provides evidence that *B. amyloliquefaciens* TL positively contributes to the health of the host, indicating its probiotic potential. Future research must focus on the mechanism of the interaction between the intestinal flora and specific host intestinal epithelial cells. The effectiveness of probiotics depends on the bacterial strain and host health status. Therefore, it is important to determine the effect of probiotics in different hosts in different health states.

## 5. Conclusions

In conclusion, the results of this study indicate that *B. amyloliquefaciens* TL alters ileal gene expression. This probiotic reduced the expression of genes involved in the inflammatory response, intestinal inflammatory factors, and receptors in the ileal tissue of 21-day-old broilers. This suggests that probiotics might reduce the damage to epithelial cells from inflammation, and thereby reduce the burden of pathogens and energy consumption caused by immune stimulation and activity, concomitantly improving broiler growth. Our study provides new insights into the role of probiotics in the diet. The findings might help optimize the use of probiotics as feed additives, instead of antibiotics, to achieve high productivity in the broiler industry.

## Figures and Tables

**Figure 1 microorganisms-09-00382-f001:**
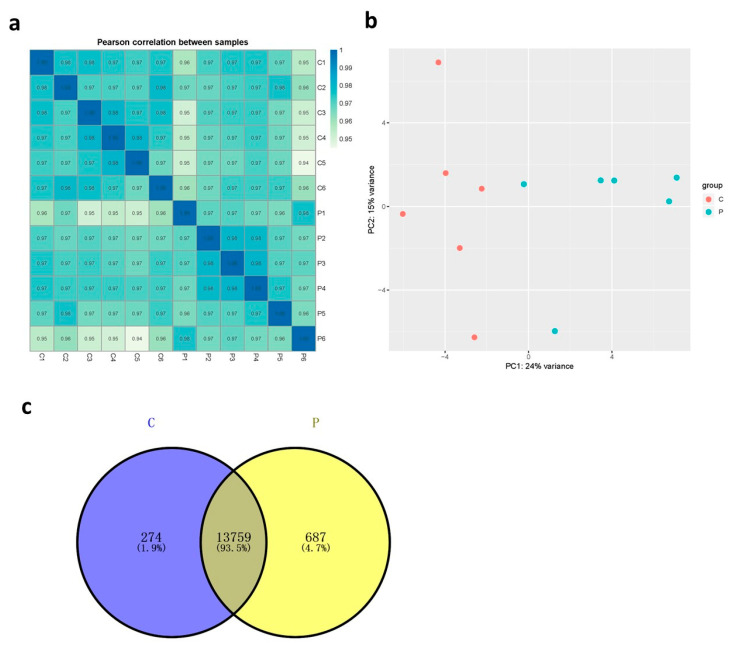
Transcriptome profiles of ileal tissues from 21-day-old broilers. (**a**) Correlation coefficient heat map between individual samples. The value is the square of the correlation coefficient of Pearson (R2) (*p* < 0.01). (**b**) Principal component analysis of transcriptomes from control group and probiotics group. (**c**) Venn diagram of the number of expressed genes detected in control group and probiotics group. “P” = *Bacillus amyloliquefaciens* TL group; “C” = control group.

**Figure 2 microorganisms-09-00382-f002:**
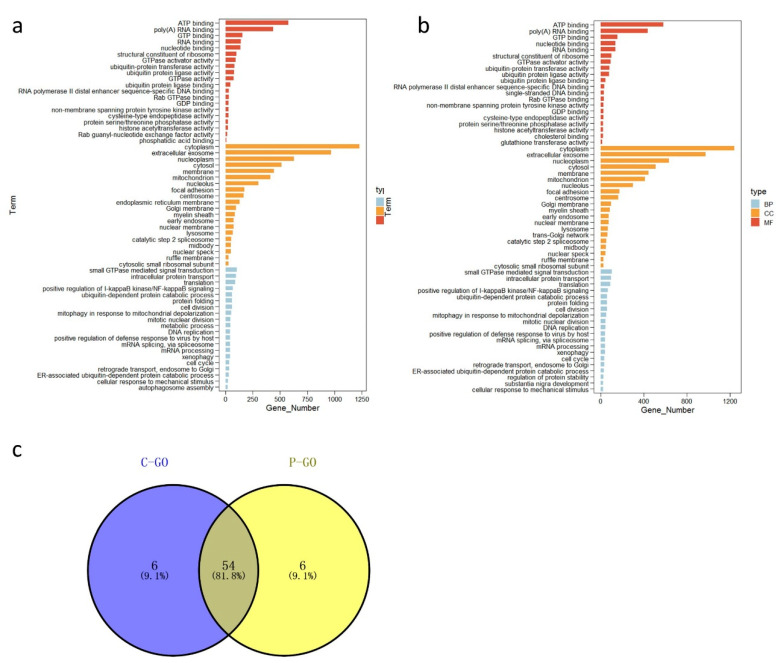
Gene Ontology (GO) terms of ileal transcriptomes. (**a**) GO enrichment analysis of the all expressed genes scattered in the C group (top 60). (**b**) GO enrichment analysis of all expressed genes scattered in the P group (top 60). (**c**) Venn diagram of the first 60 GO terms of Groups C and P. “BP” = biological process, “CC” = cellular component, “MF” = molecular function, “C” = control group, and “P” = *Bacillus amyloliquefaciens* TL group.

**Figure 3 microorganisms-09-00382-f003:**
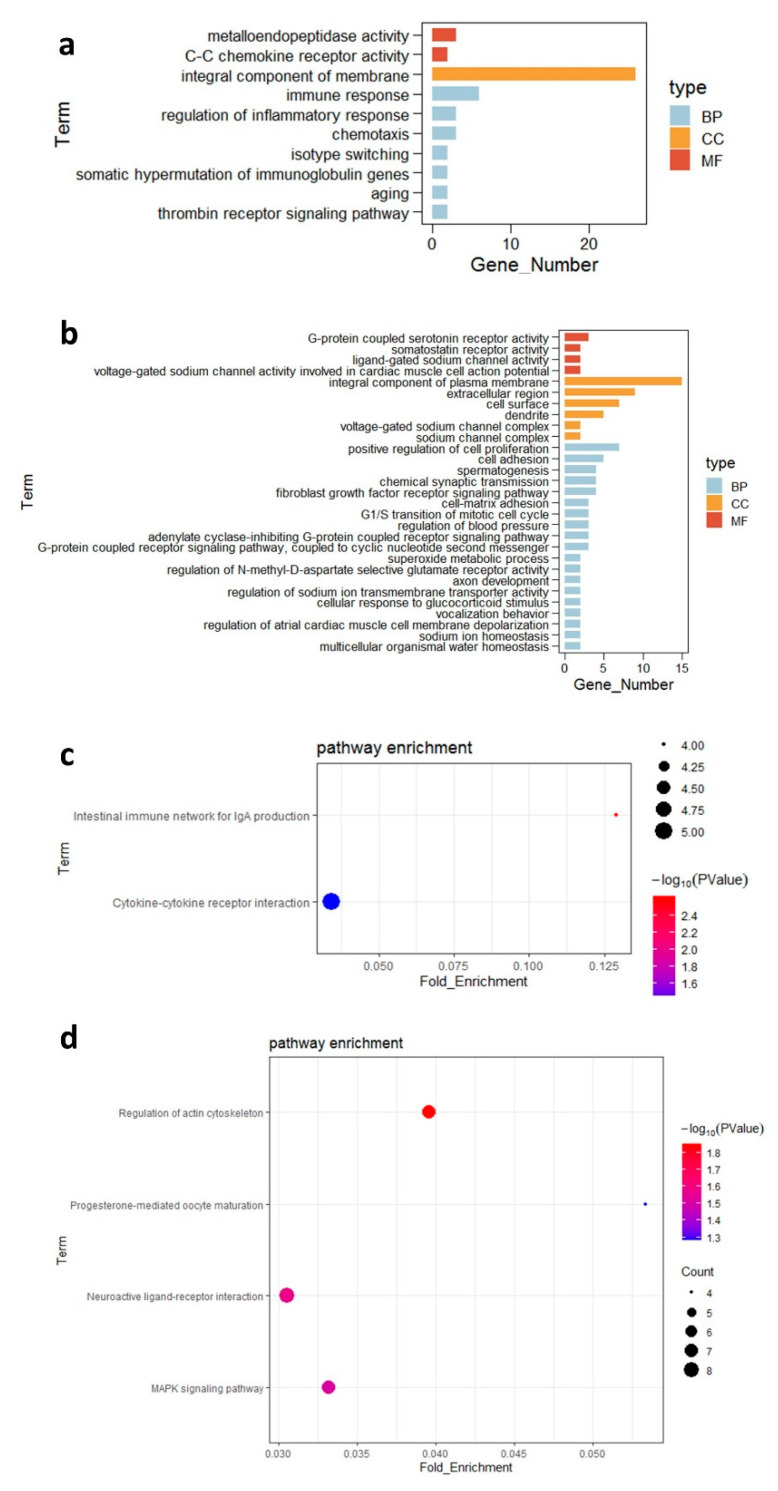
Functional enrichment analysis of uniquely expressed genes (**a**) Gene ontology (GO) enrichment analysis of the uniquely expressed genes scattered in the C group. (**b**) GO enrichment analysis of the uniquely expressed genes scattered in the P group. (**c**) Enriched Kyoto Encyclopedia of Genes and Genomes (KEGG) pathways of differentially expressed genes (DEGs) in the C group. (**d**) Enriched KEGG pathways of DEGs in the P group. “BP” = biological process, “CC” = cellular component, “MF” = molecular function, “C” = control group, and “P” = *Bacillus amyloliquefaciens* TL group.

**Figure 4 microorganisms-09-00382-f004:**
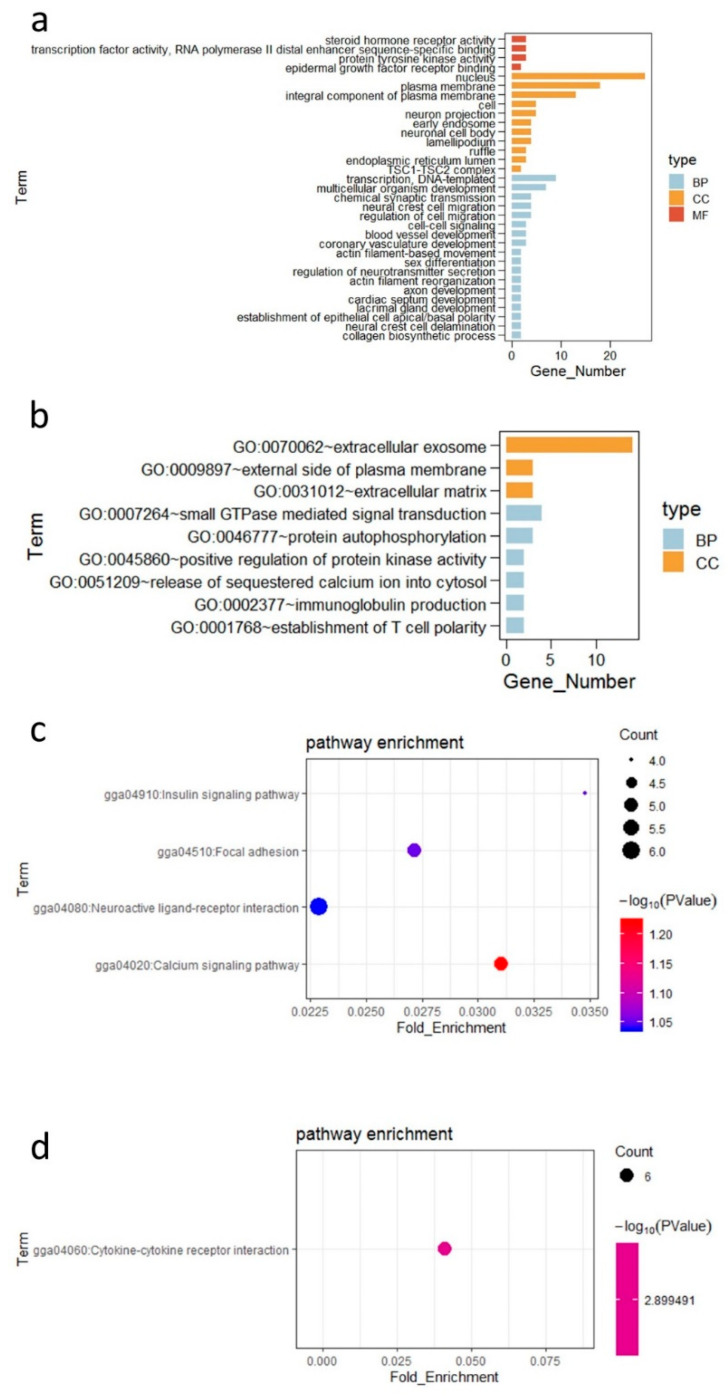
Functional enrichment analysis of differentially expressed genes (DEGs). (**a**) Gene ontology (GO) enrichment analysis of the upregulated DEGs between C group and *p* group. (**b**) GO enrichment analysis of the downregulated DEGs between C group and P group. “BP” = biological process, “CC” = cellular component, and “MF” = molecular function. (**c**) Kyoto Encyclopedia of Genes and Genomes (KEGG) pathway enrichment of upregulated DEGs. (**d**) KEGG pathway enrichment of downregulated DEGs. The vertical axis represents the name of the pathway, and the horizontal axis represents the Rich factor. The size of the point indicates the number of DEGs in the pathway, and the color of the point corresponds to a different Q value ranges. “P” = *Bacillus amyloliquefaciens* TL group; “C” = control group.

**Figure 5 microorganisms-09-00382-f005:**
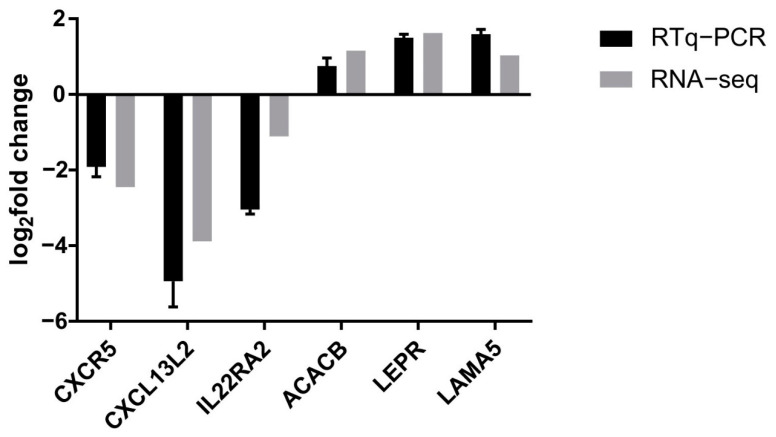
The fold expression changes in six randomly selected genes to validate the RNA-seq results.

**Figure 6 microorganisms-09-00382-f006:**
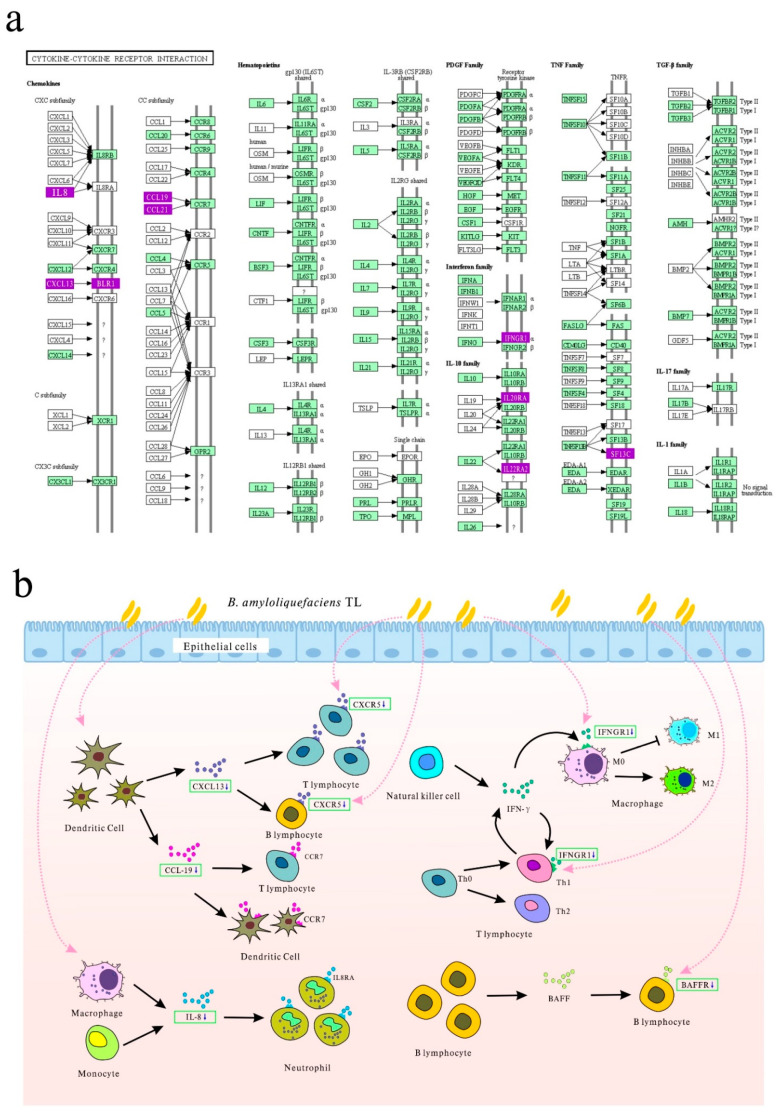
*B. amyloliquefaciens* TL acts on broiler ileum tissues and significantly downregulates immune-related genes in the “cytokine–cytokine receptor interaction” pathway. (**a**) “cytokine–cytokine receptor interaction” pathway. The genes in the green boxes represent genes expressed in birds, and the genes in the purple box represent the downregulated genes related to immune activity obtained in this study. (**b**) Schematic diagram of the function of the cytokine or cytokine receptor encoded by the downregulated gene. The interaction between the chemokine CXCL13 and its sole receptor CXCR5 guides B cells to B cell follicles and participates in the zoning of germinal centers. It also attracts T cells to B cell areas and mediates the interaction between B and T cells [66]; the interaction of BAFF and its receptor BAFFR (encoded by *TNFSF13C*) plays a central mediator in the homeostasis of mature B cell subsets, mainly through the modulation of cell survival. Unlike mammals, BAFF is also involved in early B cell development in birds [68]; the interaction of IL-8 with IL8RA, a G-protein-coupled receptor is responsible for the activation of neutrophils. An IL-8-activated neutrophil is characterized by enhanced migration, phagocytosis, superoxide generation, and granule release [60]; IFN-γ interacts with its receptor IFNGR1, activates type I macrophages, enhances the activation of both MHC-I and MHC-II molecules, and assists in antigen presentation and processing, as well as removal of intracellular pathogens, and prevents virus replication. Moreover, IFN-γ activates T-helper I-type immune responses [70]; the chemokine CCL19 interacts with its receptor CCR7 and plays a pivotal role in T cell and dendritic cell trafficking into Bursa of Fabricius and other lymphoid tissues [76]. *B. amyloliquefaciens* TL acts directly or indirectly (indicated by the pink dotted arrow in the figure) on immune cells to reduce the expression of corresponding cytokines or cytokine receptors, thereby downregulating the above-mentioned immune activity and reducing the energy consumption of immune activation, which is beneficial for the accumulation of energy in broilers.

**Table 1 microorganisms-09-00382-t001:** Primers used in this study.

Name.	Sequence (5′–3′)
*β-actin*	F: TATTGCTGCGCTCGTTGTTG
	R: TGGCCCATACCAACCATCAC
*CXCR5*	F: CCTATGACTTGAGCCTGGTGG
	R: TCCCAGCACGAACATAAGCAG
*LAMA5*	F: GCACATCCCATAACGAACGC
	R: TCAGGACGAGGGGAATTTGC
*ACACB*	F: TTCGGGACTTCAACCGTGAG
	R: GGCTGCTTAAAATCCCGCAG
*CXCL13*	F: CAGCCATCCTGGAAGCCAAC
	R: GGATCCACACAGATCCTCTCG
*LEPR*	F: AGTGCAAACATGCAAGCGAG
	R: CAGCTTGCCTTCAACCCAAC

**Table 2 microorganisms-09-00382-t002:** Effects of *Bacillus amyloliquefaciens* TL on the growth performance of broilers.

Group		Control	Probiotics	*p*-Value
	1 days	40.84 ± 1.48	39.58 ± 1.19	0.065
Average weight	7 days	147.48 ± 11.08	157.35 ± 8.58	0.053
(g)	14 days	499.75 ± 20.67	532.50 ± 24.09	0.056
	21 days	858.62 ± 30.74 ^A^	1076.12 ± 98.48 ^B^	0.000
-	35 days	1901.38 ± 76.36 ^a^	2156.12 ± 198.61 ^b^	0.017
	1–7 days	15.23 ± 1.38 ^a^	16.82 ± 1.08 ^b^	0.022
Average daily	8–14 days	50.32 ± 1.51	53.59 ± 2.33	0.074
gain (g)	15–21 days	51.27 ± 1.74 ^A^	77.66 ± 10.74 ^B^	0.000
	22–35 days	80.21 ± 3.72	83.08 ± 9.23	0.608
Feed conversion ratio		2.02	1.62	

Note: The result is expressed as Mean ± SEM (n = 60 chickens per group in a total of 120). Different capital letters between columns indicate extremely significant differences (*p* < 0.01); different lowercase letters between columns indicate significant differences (*p* < 0.05); the same lowercase letters or no letters indicate nonsignificant differences (*p* > 0.05).

**Table 3 microorganisms-09-00382-t003:** RNA-sequencing (RNA-seq) analysis in Kyoto Encyclopedia of Genes and Genomes (KEGG) Pathway.

KEGG Pathway	*p*-Value	Gene Number	Molecules
Up group			
Calcium signaling pathway	0.059866	5	*PHKA1*, *ADCY3*, *P2RX1*, *HTR6*, *ITPR3*
Focal adhesion	0.088308	5	*COL27A1*, *LAMA5*, *RAPGEF1*, *KDR*, *COL5A1*
Insulin signaling pathway	0.089046	4	*TSC2*, *PHKA1*, *ACACB*, *RAPGEF1*
Neuroactive ligand–receptor interaction	0.091555	6	*SSTR1*, *GRIK4*, *GRM2*, *CHRM4*, *P2RX1*, *HTR6*
Down group			
Cytokine–cytokine receptor interaction	0.001260	6	*TNFRSF13C*, *IFNGR1*, *CXCL13L2*, *IL20RA*, *CCL19*, *CXCR5*

## Data Availability

The data that support the findings of this study are available from the corresponding author upon reasonable request.

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
