# Peer review of "Bacillus amyloliquefaciens TL Downregulates the Ileal Expression of Genes Involved in Immune Responses in Broiler Chickens to Improve Growth Performance"

_microorganisms, 2021, doi:10.3390/microorganisms9020382_

Round 1
Reviewer 1 Report
This article is well compiled, there is a minor typo on Table 1 ,
B-action Table 1 should be B -actin
Author Response
We thank you for your thoughtful comments and suggestions. We have revised our manuscript based on these comments and suggestions and have provided a point-to-point response to these comments and suggestions (below). We hope we have addressed all of these comments and suggestions to your satisfaction. We look forward to working with you to move this manuscript closer to publication in the Microorganisms.
A point-to-point response to the comments and suggestions provided by reviwers
Reviewer 1
- β-action Table 1 should be β-actin
Answer: As indicated, we have now corrected this spelling error in line 180 in our revised manuscript. Thank you for your correction.
Reviewer 2 Report
This study characterized the effect of the probiotic: B. amyloliquefaciens TL on the ileal expression of genes involved in immune responses in broiler chickens. The authors have used RNA-Seq-based trancriptome analysis to elucidate the functional changes in the ileal tissues of broiler chicken. The study is well done and needs minor revisions; however, experimental design should be well explained and accurate statistical analysis must be added to the result section. Also, I recommend designing a concluding figure to describe proposed regulatory and molecular mechanisms of B. amyloliquefaciens TL responsible for improving intestinal inflammation and systemic immune response in broiler chickens. Here are my detailed comments: Abstract Line 16-17: “significantly higher”, add P value Introduction Line 37: “Bacillus spp. are commonly used as probiotics in the poultry industry”: is it true for global consumption? Line 43-44: Redundant, please rephrase Line 53-55: Again redundant. Please rephrase Material and methods Line 79-81: How was dose standardized? Did you previously perform a dose dependent study? Any positive control? Any established probiotics used in this study? Line 92: Did you sample day 1 to collect baseline data? Line 93: Rationale to choose day 21? Are 6 biological replicates enough for an in vivo statistical study? Were there any mortalities in two groups? Line 105: What does qualified samples mean? Please include total n at each step. Line 154: Table1: please edit “β-action”? Line 159: “Experiments were performed in triplicate”? Biological or technical? Results Please add absolute P values while describing significant differences. Line 164: “The average daily weight gain of the probiotic group was higher than that of the control group at 0–7days”, P value? Table 2: Add another column to include absolute P values. Line 177: “The transcriptome data from the 12 ileum samples” sampling with total n is not included in methods. Please add total n in method section. Line 181: “0.972 and 0.965, respectively, indicating that the sequencing data were suitable for the subsequent analyses”. Add P value. Include R value to show a positive or negative correlation. Discussion Please add a concluding figure/pathway to describe how B. amyloliquefaciens TL altered ileal gene expression to improve immunity in chickens.
Author Response
We thank you for your thoughtful comments and suggestions. We have revised our manuscript based on these comments and suggestions and have provided a point-to-point response to these comments and suggestions (below). We hope we have addressed all of these comments and suggestions to your satisfaction. We look forward to working with you to move this manuscript closer to publication in the Microorganisms.
Reviewer 2
- experimental design should be well explained and accurate statistical analysis must be added to the result section.
Answer: as indicated, we have now provided detailed explanations of our experimental design (lines 82-88, 96-101, 103-112 and 125-128) and statistical analysis (lines 191 and 193) in our revised manuscript.
- Also, I recommend designing a concluding figure to describe proposed regulatory and molecular mechanisms of B. amyloliquefaciens TL responsible for improving intestinal inflammation and systemic immune response in broiler chickens.
Answer: As suggested, we have now added a concluding figure to describe the proposed regulatory and molecular mechanisms of B. amyloliquefaciens in lines 500-523 in our revised manuscript.
Detailed comments
- Abstract Line 16-17: “significantly higher”, add P value Introduction.
Answer: The genes encoding pyruvate kinase, prothymosin-α and heat stress proteins were not selected by P-value, but based on the condition of FPKM > 500. Therefore, we have now added explained this situation in our revised manuscript in lines 18-19.
- Line 37: “Bacillus spp. are commonly used as probiotics in the poultry industry”: is it true for global consumption?
Answer: We believe that Bacillus spp. is globally consumed because Bacillus spp. have been applied in many countries, including China, America, Germany and South Africa, etc., as supported by various publications. To clarify this issue, we have revised this sentence in our revised manuscript. Specifically, the revised sentence now reads as "It has been demonstrated that dietary supplementation of Bacillus spp. can alter the intestinal microbiota to help prevent infectious diseases and increase productivity ". We have also cited a few additional references to support this statement in lines 38-40.
- Line 43-44: Redundant, please rephrase.
Answer: According to reviewer’s comment, we have merged the two sentences from lines 43-46 into one sentence to avoid redundancy.
- Line 53-55: Again redundant.
Answer: As indicated, we have now deleted the sentence in line 53 to avoid this redundancy.
- Please rephrase Material and methods Line 79-81: How was dose standardized? Did you previously perform a dose dependent study? Any positive control? Any established probiotics used in this study?
Answer: As indicated, we have now revised this portion in our revised manuscript (lines 82-87). Specifically, the B. amyloliquefaciens TL strain used in this experiment was provided by Hubei Huada Real Technology Co. (Wuhan City, Hubei Province, China). It is a commercial probiotic powder that has been put on the market, with an effective viable number of 2.0 × 1010 CFU/g. The concentration of our feed additives is 200 g/ton. In the early stage, we conducted 100 g/ton, 200 g/ton, and 400g /ton measurement dependent studies in the market. Our results showed that 200 g/ton generated the highest economic benefit. Positive control was performed in this experiment, showing that the addition of 200 g/ton is similar to the effect of 50 g/ton chlortetracycline. We have now provided a new reference to support these statement (Hong et al. 2019).
Hong, Y.; Cheng, Y.; Li, Y.; Li, X.; Zhou, Z.; Shi, D.; Li, Z.; Xiao, Y. Preliminary Study on the Effect of Bacillus amyloliquefaciens TL on Cecal Bacterial Community Structure of Broiler Chickens. Biomed Res Int 2019, 2019, 5431354, doi:10.1155/2019/5431354.
- Did you sample day 1 to collect baseline data?
Answer: We only collected 21-day-old samples and did not collect 1-day-old samples. This is because: (1) we focus on whether the addition of Bacillus amyloliquefaciens TL in the diet will increase the weight of broilers, and the blank group of broilers fed with the basic diet will be used as a control group. (2) As the age changes, the intestinal transcriptome of the chicken will change greatly. The ileal transcriptome of 1-day-old chickens is expected to be similar (not significantly different) in the two treatment groups, which were not treated and were randomly grouped.
- Rationale to choose day 21? Are 6 biological replicates enough for an in vivo statistical study? Were there any mortalities in two groups?
Answer: The growth-promoting effect of B. amyloliquefaciens TL in broilers was particularly evident at 14–21 days of age. The results of the present study suggest that the growth-promoting effect of B. amyloliquefaciens TL is stronger during the broiler brooding period (1–21 days) than the fattening period (22–35 days). Therefore, we select the 21-day-old samples of two different treatment groups for comparison, which can clearly show the difference between them. These explanations are already mentioned in the discussion (lines 327-332).
The way we select samples: select the individuals in the four pens with body weight close to the average body weight of each pen. In order to expand the number of biological replicates, we select the two birds with body weight closest to the average weight of each group from all individuals in each group. We selected these representative individuals to sequence the gut transcriptome and used these representative data to predict the entire population. In addition, in this experiment, the individual differences within each group are small.
During the experiment, two birds died in each of the experimental group and the probiotic group due to weakness. When the broiler was dead, the weight of the dead broiler and the remaining feed weight were recorded to correct the growth performance data. This information has now been provided in Section 2.1 of Materials and Methods (Lines 98-101).
- What does qualified samples mean? Please include total n at each step.
Answer: The qualified samples represent the bands that passed quality control standards. Specifically, the target band of RNA electrophoresis gel is bright and clear, there is no diffusion area in the swimming lane, and no protein and DNA contamination; RNA Integrity Number (RIN) value is close to 10; 28S/18S is greater than or equal to 1.5; 1.7 <OD260/280<2.0; OD260/230 ratio is 2.5. This information has now been provided in our revised manuscript (lines 125-128). In addition, total n has been added in each step (lines 79, 97, 104, 131, 198, 225, 244, and 271).
- Line 154: Table1: please edit “β-action”?
Answer: Thank you for your correction. We have edited "action" to "actin" on line 180.
- Line 159: “Experiments were performed in triplicate”? Biological or technical? Results Please add absolute P values while describing significant differences.
Answer: qRT-PCR experiments were performed in three technical replicates. We have now moved this sentence to Materials and Methods 2.9 (Lines 175-176). We have also added P-values on lines 191, 271, and 314, in our revised manuscript.
- Line 164: “The average daily weight gain of the probiotic group was higher than that of the control group at 0–7days”, P value? Table 2: Add another column to include absolute P values.
Answer: According to your comment, the P value is added in row 191, and a new column is added to Table 2 (row 196).
- Line 177: “The transcriptome data from the 12 ileum samples” sampling with total n is not included in methods. Please add total n in method section.
Answer: According to your comment, a description of the total n has been added in each step (lines 79, 97, 104, 131, 198, 225, 244, and 271) in our revised manuscript.
- Line 181: “0.972 and 0.965, respectively, indicating that the sequencing data were suitable for the subsequent analyses”. Add P value. Include R value to show a positive or negative correlation.
Answer: We performed sample correlation testing in bioinformatics analysis in order to visualize the similarity or correlation of RNA-seq data between samples. The correlation of gene expression levels between samples is an important indicator for testing the reliability of the experiment and whether the sample selection is reasonable. The closer the correlation coefficient is to 1, the higher the similarity of the expression patterns between samples. The purpose of this step is only to visualize the relationship between samples and the degree of difference, and to provide a test for subsequent analysis. Therefore, it is not necessary to explore the positive and negative correlations of data between samples. In addition, as we indicated in the manuscript that R2 = 0.972 and R2 = 0.965 indicate the least correlations between the control group and the probiotic group, indicating that the intrasample correlation of the two groups is very high. In order to clarify this confusion, we have now revised these descriptions to “the Pearson correlation (R2) of the biological repeats in the control and probiotic groups was high, indicating that the sequencing data were suitable for the subsequent analyses.” in line 209 in our revised manuscript.
- Discussion Please add a concluding figure/pathway to describe how B. amyloliquefaciens TL altered ileal gene expression to improve immunity in chickens.
Answer: As indicated, we have now provided a concluding figure in lines 500-523 in our revised manuscript.
Reviewer 3 Report
This study elucidated the effects of the probiotic food supplementation, Bacillus amyloliquefaciens TL, on ileal gene expression by using RNA-seq based transcriptome analysis in 21-day old broiler chickens. The authors found that the B. amyloliquefaciens supplementation downregulated the expression of inflammatory factors in ileum, which helped the broiler chickens growth performance. It was overall well-organized and well-written, but some minor concerns were raised.
- The title should be changed. The title should cover overall contents and deliver the main point of the study. The main point of the study ws that B amyloliquefaciens TL downregulated the genes of the immune responses to improve the growth performance of the broiler chickens, but the present title might be misunderstood, like the probiotic supplementation helped to increase the expression of immune response genes. Please clarify the title correctly.
- The basic information of selected 6 chickens/group is needed. The authors collected the BW, water and feed consumption data from total 120 chickens (60 chickens/group), and then selected only 6 chickens/group to analyze the transcriptome. Statistically, it might be critical how to select small numbers of subjects among the large group. The author described that the 6 chickens/group were selected randomly, but the basic information of each samples (BW, growth rate, food consumption, from which pens, etc.) should be provided to convince the transcriptome data.
- And the sample number (only 6 chickens/group) seemed too low. The authors need to provide the supporting description of the small sample number in the methods or discussion section.
Author Response
We thank you for your thoughtful comments and suggestions. We have revised our manuscript based on these comments and suggestions and have provided a point-to-point response to these comments and suggestions (below). We hope we have addressed all of these comments and suggestions to your satisfaction. We look forward to working with you to move this manuscript closer to publication in the Microorganisms.
Reviewer 3
- The title should be changed. The title should cover overall contents and deliver the main point of the study. The main point of the study ws that B amyloliquefaciens TL downregulated the genes of the immune responses to improve the growth performance of the broiler chickens, but the present title might be misunderstood, like the probiotic supplementation helped to increase the expression of immune response genes. Please clarify the title correctly.
Answer: According to your comment, the title has been changed to "Bacillus amyloliquefaciens TL downregulates the ileal expression of genes involved in immune responses in broiler chickens to improve growth performance".
- The basic information of selected 6 chickens/group is needed. The authors collected the BW, water and feed consumption data from total 120 chickens (60 chickens/group), and then selected only 6 chickens/group to analyze the transcriptome. Statistically, it might be critical how to select small numbers of subjects among the large group. The author described that the 6 chickens/group were selected randomly, but the basic information of each samples (BW, growth rate, food consumption, from which pens, etc.) should be provided to convince the transcriptome data.
Answer: We agree with this reviewer. During the animal experiment, we recorded body weight and remaining feed weight in units of pen. The weight information and column numbers of the sampled individuals were recorded only on the day of sampling, and the information has been added to the supplementary table S1.
The way we select samples: select the individuals in the four pens with body weight close to the average body weight of each pen. In order to expand the number of biological replicates, we select the two birds with body weight closest to the average weight of each group from all individuals in each group. We selected these representative individuals to sequence the gut transcriptome and used these representative data to predict the entire population. This information has now been provided in our revised manuscript (103-111).
- And the sample number (only 6 chickens/group) seemed too low. The authors need to provide the supporting description of the small sample number in the methods or discussion section.
Answer: Based on your comment, we have provided the following information in Method 2.2 (lines 101-108): "Six chickens were sampled from the control and probiotic groups on day 21 (n = 6 simples per group, i..e, in a total of 12 samples) . The method of selecting samples is as follows: the individuals in the four pens with the body weight closest to the average body weight of each pen were chosen for transcriptome analysis. In order to expand the number of the biological replicates, we selected the two birds with body weight closest to the average body weight of each group in each group. We selected the two representative individuals to conduct the gut transcriptome analysis, and used these representative data to predict the entire population.”
Round 2
Reviewer 3 Report
All the concerns raised were appropriately addressed by the authors. The revised manuscript now seems acceptable for publication.